# Childhood vaccination trends among the Maasai nomadic pastoralists: Insights from a community-based vaccine registry in Kenya

Julius Nyerere Odhiambo[1,2]*, Carrie B. Dolan[1,2], Evelyn Thompson[2], Katie O'Neill[2], John Sankok[3], Rose Kimani[3]

**1** Department of Health Sciences, William and Mary, Williamsburg, Virginia, United States of America,
**2** Ignite Global Health Research Lab, Global Research Institute, William and Mary, Williamsburg, Virginia, United States of America, **3** Community Health Partners, Narok, Kenya

* jnodhiambo@wm.edu

## Abstract

Inequities in vaccination timeliness and coverage contribute to disparities in childhood health and survival. Regular, reliable estimates are needed to take programmatic action and track progress towards initiatives such as the Immunization Agenda 2030. This study assessed the timeliness, coverage, and drop-out rates of reported immunization data from a community healthcare registry. We retrospectively reviewed vaccination records of 8487 children across 176 villages. The proportion of children receiving early, timely and delayed vaccination was computed by vaccine, village and year. Coverage of each vaccine was calculated as the number of reported doses divided by the number of children who received Bacillus Calmette-Guerin (BCG), a birth dose serving as the service-based denominator. Vaccine dropout by year was estimated as the proportion of children who received the first dose of a vaccine but did not receive the subsequent dose. For multi-dose vaccines, on-time vaccination rates were highest for the first dose but declined with subsequent doses. The largest declines between the first and third doses were observed in DPT (29.07%), Pneumococcal Conjugate Vaccine (28.84%), and Oral Polio Vaccine (28.79%). The Measles-Rubella vaccine had the highest dropout rate (64.66%) between its two doses, largely due to delays in administering the second dose at 18 months. Overall, vaccination coverage steadily declined from mid-2020 to 2022, with proximity to healthcare facilities strongly linked to higher coverage and lower dropout rates. The study confirmed that community level estimates were significantly below the national immunization targets. Understanding factors affecting coverage, timeliness and dropout rates at this level is important for building a strong and sustainable vaccine ecosystem for hard-to-reach communities.

## Background

Administering timely vaccinations within a child's first year of life is one of the most cost-effective methods for reducing the burden of vaccine-preventable diseases (VPDs). Since its inception in 1974, the expanded program on immunization (EPI) has significantly led to a decline in the incidence of, and mortality from childhood vaccine preventable diseases [1]. Through the program,

**Data availability statement:** All relevant data is within the manuscript and its Supporting Information files.

**Funding:** The authors received no specific funding for this work.

**Competing interests:** The authors have declared that no competing interests exist.

substantial strides have been made in reducing child mortality, with vaccination playing a crucial role in reducing under-five deaths from 93 per 1000 live births in 1990 to 38 per 1000 livebirths in 2021. Concerningly, in 2022, 20.5 million children missed one or more doses of the diphtheria, pertussis, and tetanus (DPT) vaccine delivered through routine immunization services, underscoring the need for catch-up efforts, recovery initiatives and health system strengthening [2].

Routine immunization is a fundamental pillar of primary healthcare with a significant return on investment (ROI). Studies estimate a ROI of over 16 dollars for every dollar spent, rising to 48 dollars when considering broader benefits like improved public health and reduced healthcare costs [3]. However, missed opportunities for immunizations continue to impact children'schildren health, education, and future productivity with sub-Saharan Africa being disproportionately affected with 1 in 14 children likely to die before their fifth birthday [4]. The Immunization Agenda 2030: A Global Strategy to Leave No One Behind, seeks to extend immunization services to zero-dose and under immunized children. As a signatory of the WHO's Global Vaccine Action Plan, Kenya committed to fully immunizing 90% of all children by 2020, targeting at least 80% coverage in each sub-national region. However, the COVID-19 pandemic disrupted immunization services, resulting in more children remaining unvaccinated by 2020 [5,6]. This disruption not only reversed decades of progress but also exacerbated existing inequities in vaccine access, particularly for children in communities affected by conflict, underserved remote areas, and informal settings [6]. In these settings, children continue to face multiple deprivations, including limited access to basic health services, leaving them vulnerable to outbreaks of vaccine-preventable diseases.

In tracking progress towards the achievement of Sustainable Development Goals (SDGs) by 2030, which has an overarching aim of "leaving no one behind," the global health and development community has increasingly recognized the importance of precise geographical data in estimating health indicators [7]. However, vaccination metrics are often reported at national or continental levels, primarily due to administrative convenience, operational constraints, and high data collection cost, which limits the production of spatially detailed estimates. Moreover, differences in observation year and the sampling errors often obscure critical subnational variations in vaccine uptake and timeliness, masking epidemiologically significant disparities that can inform targeted interventions [8–10]. As more counties in Kenya transition from paper-based registries to individual-level electronic records, monitoring vaccination metrics at subnational levels remains critical for implementing vaccine catch-up strategies and addressing immunity gaps caused by disruptions in immunization programs [11,12].

Few empirical studies have explored the link between suboptimal vaccine coverage and immunity gaps to pinpoint regions and communities in need of targeted catch-up efforts [13,14]. This is important for Kenya, where nomadic communities, such as the Maasai, face disrupted access to essential healthcare due to their frequent movement in search of water and pasture [11,15]. Historically, the success of EPI programs/vaccination success has typically been assessed exclusively by vaccination coverage rates, which measures the uptake of vaccines, but fail to capture the unique challenges faced by mobile communities and immunization programs. Furthermore, this approach neglects vaccine timeliness, a critical quality dimension at both the individual programmatic levels. At the individual level, vaccinations administered too early may lead to suboptimal immune responses due to interference with maternal antibodies. Conversely, delayed vaccinations can increase children's vulnerability to vaccine-preventable diseases (VPDs) like pertussis and measles, which are often encountered in the first year of life [16,17]. At the programmatic level, early or delayed vaccinations can signal issues with vaccine delivery and access.

The vaccination schedule in Kenya relies on repeated co-administration of multiple vaccine antigens to establish immunity and reduce the risk of preventable infectious diseases. While childhood vaccination coverage has improved in recent years, with most vaccines now exceeding the 80% target, this metric often masks variability in vaccination timelines and drop-out

rates, especially in urban informal settlements and among communities in remote or hard-to-reach areas. The Maasai, with their migratory lifestyle, face unique challenges in accessing consistent care and building strong relationships with healthcare providers, which contributes to lower trust in essential services like childhood vaccinations. Moreover, limited data on the medical care received makes it difficult to identify and address their specific health needs, hindering the development of effective, tailored interventions. To address these gaps, we aimed to assess vaccination coverage, timeliness, and drop-out rates within an underserved Maasai population using data from a community-based vaccine registry. Community-level data offers valuable insights that can inform efforts to improve access, availability, and demand for immunization services, particularly for zero-dose children.

## Methods

### Study area

The study gathered immunization data retrospectively from four primary healthcare clinics located in Aitong, Ewaso-Ngiro, Mara-Rianta, and Talek. These clinics provide vaccination services as per Kenya's EPI schedule (Fig 1). These clinics began documenting their child

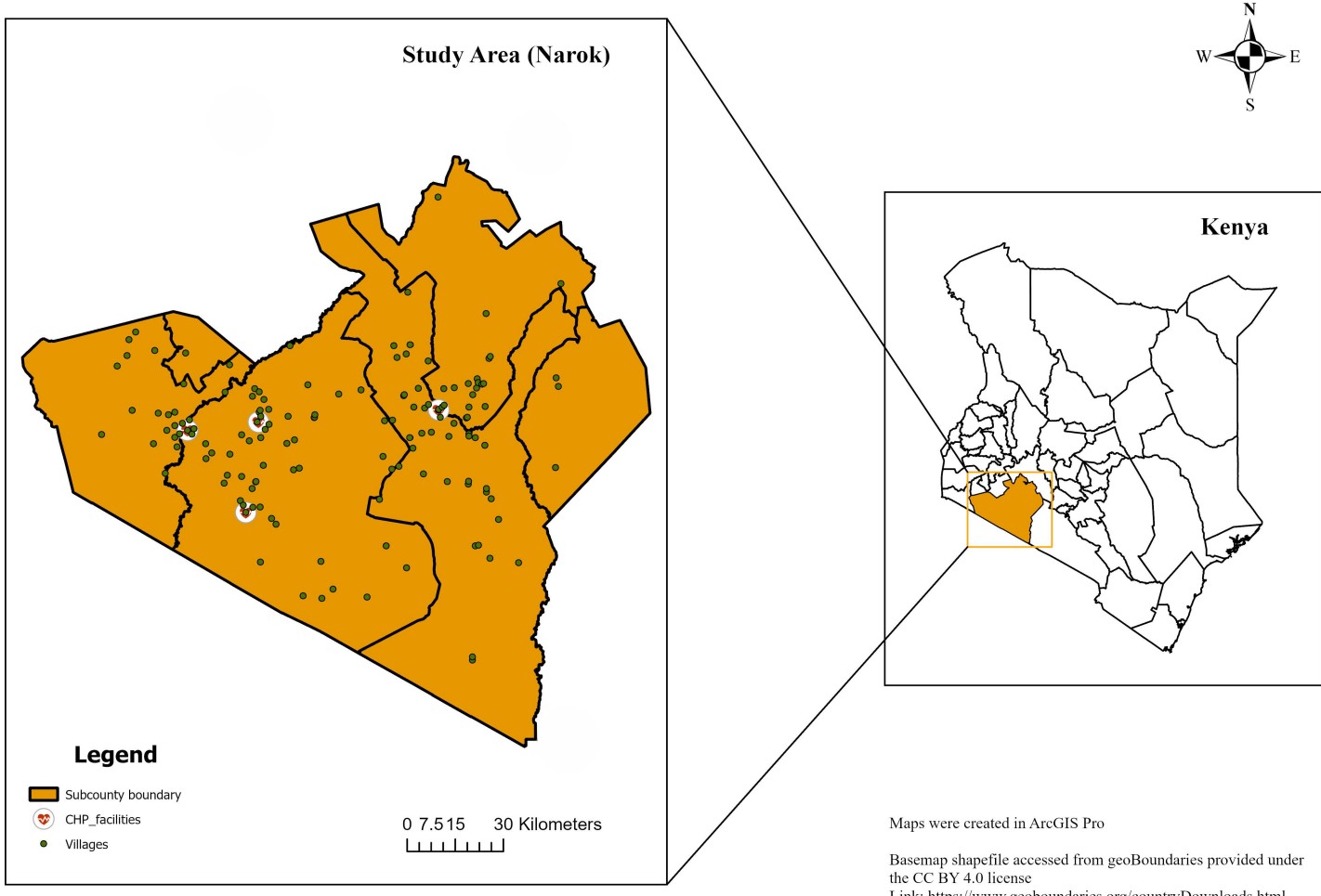

**Fig 1. Study Area.** Maps were created in ArcGIS Pro. Basemap shapefile accessed from geoBoundaries provided under the CC BY 4.0 license [18]. The shapefiles are publicly accessible. https://www.geoboundaries.org/countryDownloads.html.

immunization records at different points in time. Talek commenced its record-keeping in 2018, while the other three clinics initiated theirs in 2016. (S2 Table).

## Data sources

Data on vaccinations was extracted from each facility's immunization permanent register (MOH 510) booklet and the mother and child booklet (MOH 216), from 21 July 2023 to 19 June 2024. However, in case of discrepancies between the two sources, the MOH 510 booklet was regarded as the primary source of truth. Each child was assigned a unique serial number, which was used to longitudinally track their immunization status. Data was also collected for each vaccine with the child's corresponding vaccination date, gender, village and term of birth. Children residing in the study area were eligible for enrollment within 1 month of their first birthday. New children could enter the cohort either by birth or in-migration to the study area. They could exit the study population by either out-migration or death. Children were followed up annually through a working partnership between CHP and the College of William and Mary.

## Measures

**Timeliness** by vaccine dose was calculated by subtracting the date of birth from the date of vaccination. The age in days were then compared to WHO and Kenyan Ministry of Health recommended immunization schedules. Vaccine doses were classified as "early" if administered before the recommended age, "on-time" if given within the recommended timeframe, and "delayed" if administered after the specified recommended age and "unknown" if the date for vaccination was missing or erroneously captured. These vaccines schedules included the initial administration of Bacillus-Calmette-Guérin (BCG) at birth. Rotarix (rotavirus vaccine) at 6 and 10 weeks, measles and rubella vaccine (MR) at 9 and 18 months. At weeks 6, 10 and 14, children receive Pentavalent (Penta, a combination vaccine comprising five antigens: diphtheria- -pertussis-tetanus (DPT), hepatitis B (HBV), and Haemophilus influenzae type b (Hib), Pneumococcal Conjugate Vaccine (PCV), and oral polio vaccine (OPV). Inactivated polio vaccine (IPV) is co-administered with OPV 3 at 14 weeks (Table 1). Untimely vaccinations were defined as those vaccines received outside the specified interval.

For multi-dose vaccines, dropout rates were calculated by subtracting the number of children who received the subsequent dose (e.g., dose 2 or dose 3) from those who received the preceding dose (e.g., dose 1 or dose 2), and dividing this difference by the number of children who initially received the preceding dose.

**Vaccine antigen coverage** was computed using a service-based denominator for all the children presenting at the facility and was highly dependent on the accuracy of reporting by CHP health facilities. The denominator was the count of all children who had received the

**Table 1. WHO and Kenyan Ministry of Health Vaccination Timelines.**

| Recommended age | Vaccine | Early | Delayed |
|---|---|---|---|
| Birth | BCG | | > 28 days |
| 6 weeks | DPT-HepB-Hib 1 PCV 1, OPV 1, Rotavirus 1 | < 42 days | > 70 days |
| 10 weeks | DPT-HepB-Hib 2 PCV 2, OPV 2, Rotavirus 2 | < 70 days | > 98 days |
| 14 weeks | DPT-HepB-Hib 3 PCV 3, OPV 3, IPV | < 98 days | > 126 days |
| 9 months | Measles and Rubella 1 | < 270 days | > 540 days |
| 18 months | Measles and Rubella 2 | < 540 days | |

Source: Kenya National Immunization Policy Guidelines [23].

BCG vaccine and were expected to return for follow-up immunizations [19–21]. To reduce the potential underestimation of coverage arising from the inclusion of ineligible children, the denominator was determined by considering vaccines administered at specific intervals aligned with a child's eligibility for each dose. For each vaccine dose, the data extracted were summarized by the corresponding village, which was the geographic unit of our analysis. Data analyses were done using R version 4.3.2 (R Core Team) [22].

### Ethical approval

The study was limited to previously collected and fully anonymized secondary data. Ethical approval was received from the College of William and Mary, reference number: PHSC-2022-09-21-15860 and locally from Jaramogi Oginga Odinga Teaching and Referral Hospital, reference number: ISERC/JOOTRH/432/23.

## Results

### Vaccination coverage

The analysis included 8487 vaccination records for children from 176 villages spread across different sub-counties in Narok. Between 2017 and 2020, the coverage for the first dose of DPT, OPV and PCV was above the GVAP target of 90%. However, this trend was not observed for the first dose of the MR vaccine, which had the lowest coverage rates throughout the study period. Coverage for the second and third doses of DPT, OPV, and PCV showed a steady increase from 2017 to 2020 but remained below the GVAP targets (S1 Fig). Across all vaccines, there was a gradual decline in coverage rates from mid-2020 to 2022 (Fig 2).

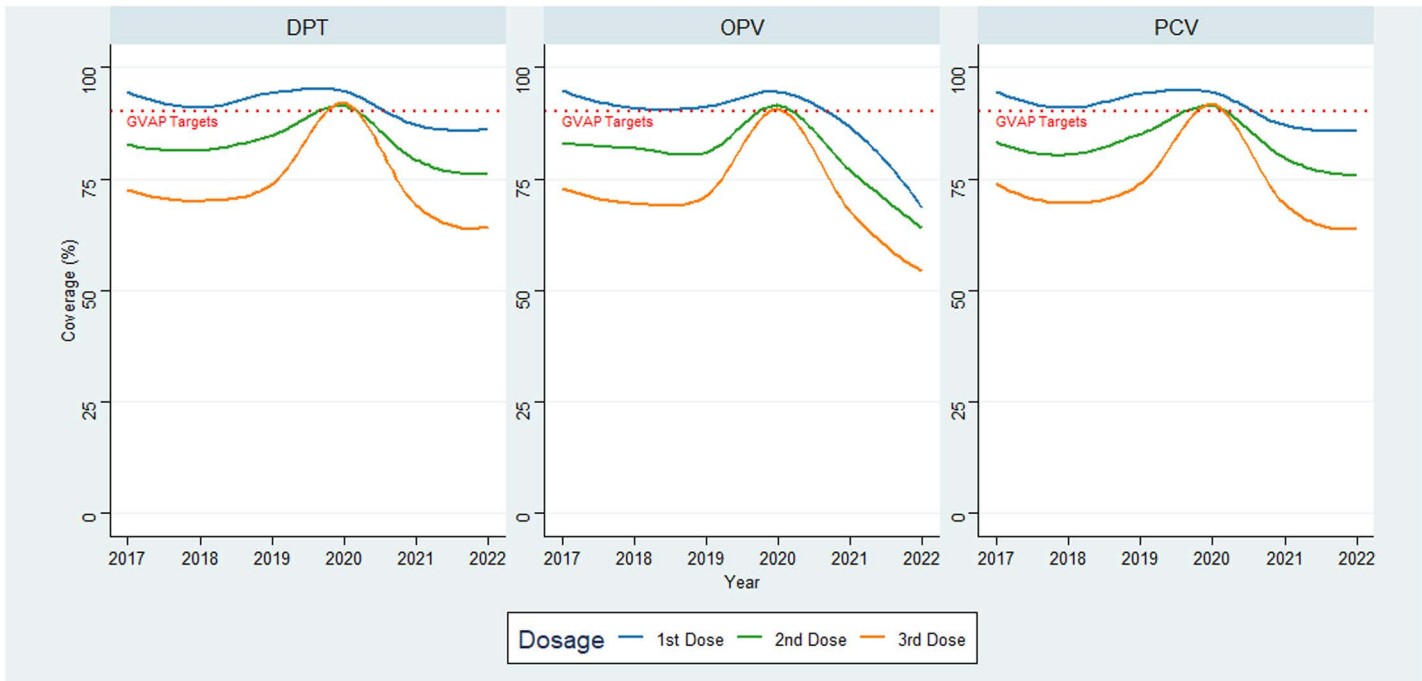

**Fig 2. Vaccine coverage by year.** The red dotted line indicates the Global Vaccine Plan (GVAP) vaccination coverage target.

Maps were created in ArcGIS Pro. Basemap shapefile accessed from geoBoundaries provided under the CC BY 4.0 license [18]. The shapefiles are publicly accessible. https://www.geoboundaries.org/countryDownloads.html

Fig 3 shows the differences in coverage for the first and third dose of DPT across the villages in the study area. Specifically, the red color indicates a coverage below 80%, blue indicates coverage between 80–90% and the yellow color indicates coverage that is above 90%. A high coverage of the first dose of the diphtheria-tetanus-pertussis-containing vaccine implies good access to primary health care facilities, whereas a high proportion of zero-dose children suggests either low coverage or a lack of acceptance of vaccination services [24]. From the maps it is evident facility location played a key role in coverage, with populations closer to

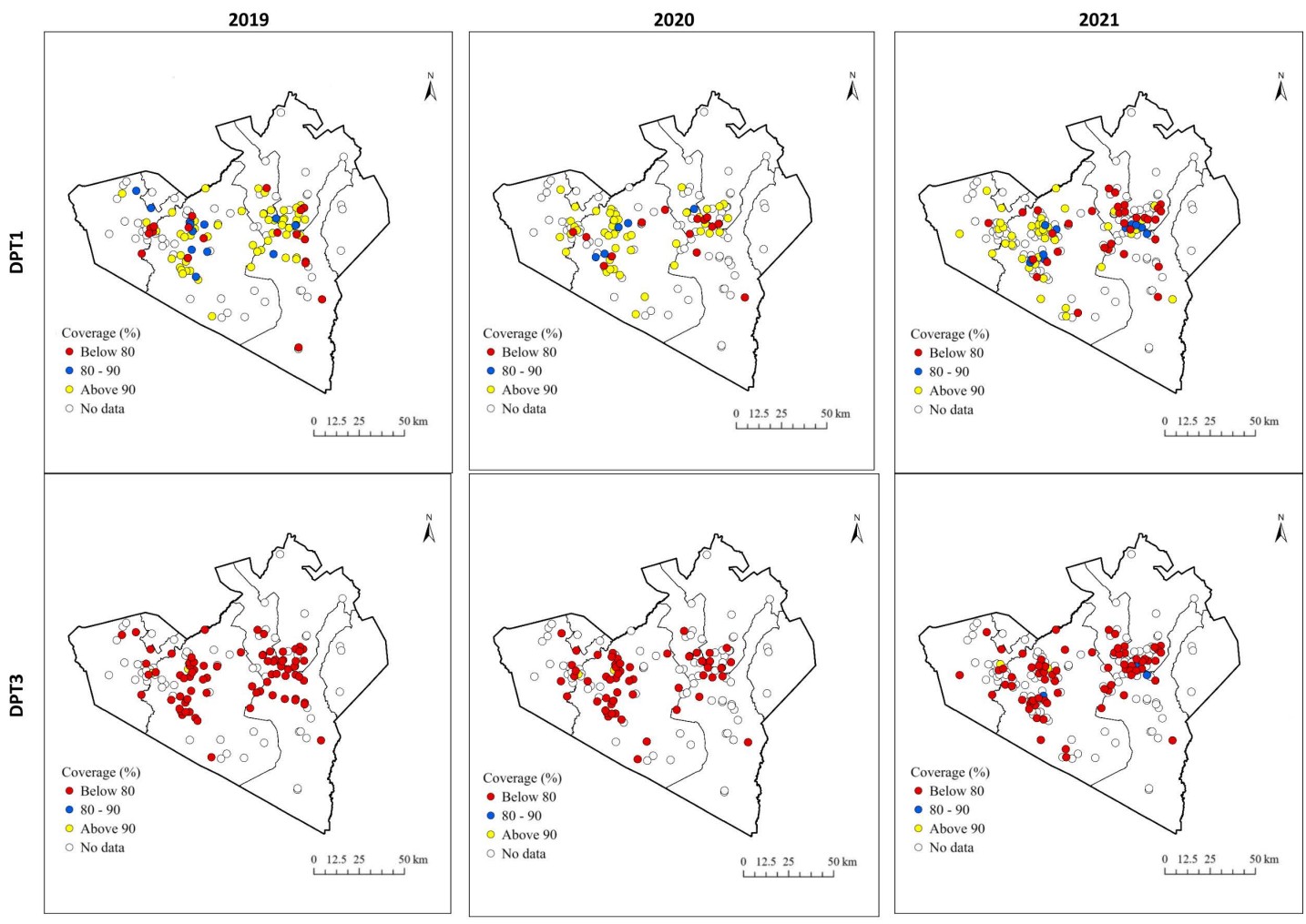

Maps were created in ArcGIS Pro

Basemap shapefile accessed from geoBoundaries provided under the CC BY 4.0 license

Link: https://www.geoboundaries.org/countryDownloads.html

Runfola, D. et al. (2020) geoBoundaries: A global database of political administrative boundaries. PLoS ONE 15(4): e0231866. https://doi.org/10.1371/journal.pone.0231866

**Fig 3. Spatial distribution of the observed vaccine DPT (1,3) coverage by village.**

CHP facilities exhibiting higher coverage (S3 Table). Overall, coverage levels declined with subsequent dosage.

### Timeliness of vaccines

For the multi-dose vaccines, the proportion of children receiving on-time vaccinations was high for the first dose but decreased for subsequent doses. For example, the highest decline between the first and third doses was observed in DPT at 29.07%, PCV at 28.84%, and OPV at 28.79%. A visual representation of this trend can be seen in Fig 4, illustrating the decline in on-time vaccinations with subsequent doses and increasing distance from the clinic.

Delayed vaccinations varied across doses and vaccines. MR's first dose had the highest delay rate at 31.02%, followed by BCG (24.03%) and the third dose of OPV (22.65%). DPT and PCV also had notable delays at 21.90% and 21.59%, respectively. Additionally, the proportion of children with unknown vaccination status increased with greater distance from the clinic across all vaccines, highlighting potential challenges in tracking and monitoring vaccination schedules in more remote areas (S1 Table).

### Dropout rates

Table 2 shows the dropout rates by dose of multidose vaccines, along with their corresponding 95% confidence intervals (CI). The Oral Poliovirus Vaccine (OPV), Diphtheria, Pertussis, Tetanus (DPT), and Pneumococcal Conjugate Vaccine (PCV) share a consistent pattern of incremental dropout rates across successive doses. Intriguingly, the Rotavirus (Rota) vaccine, administered concurrently with three-dose vaccines, demonstrates a comparatively higher dropout rate of 14.70% between its first and second dosage. Overall, the Measles and Rubella (MR) vaccine had the highest dropout rate of 64.66% between its two doses, attributable to a notable delay in the second dose administration at 18 months (S4 Table). These findings shed light on the retention challenges within vaccination schedules, emphasizing the importance of addressing barriers to completion for optimal vaccine effectiveness.

## Discussion

With more infants starting life in contact with the health system, reducing the burden of zero-dose children and identifying missed communities for targeted and tailored vaccinations remains critical. Additionally, as Kenya transitions from reliance on external support to independently sustaining its primary vaccination programs, community-based health registries play a vital role in tracking vaccination trends and guiding micro-planning initiatives, especially for underserved migrating populations like the Maasai.

Our results revealed village-level disparities in vaccine uptake and a declining dose coverage rate for all the vaccines from 2020. This trend may stem from multiple factors, many of which were exacerbated or directly caused by the Covid-19 pandemic. Globally, the United Nations Children's Fund (UNICEF) estimated that approximately 67 million children missed out on essential vaccines between 2019 and 2021, primarily during the peak of the pandemic, with coverage levels dropping in 112 countries. A 2020 multi-country analysis revealed that 13 out of 15 African countries showed a decline in the monthly number of vaccine doses administered for various antigens, with six countries having a reduction of over 10% [25]. In Kenya approximately 600,000 children missed out on life-saving routine vaccines due to intense demands on health systems and the diversion of resources toward Covid-19 vaccination efforts. As a result, a substantial and growing immunity gap persists among populations disproportionately affected by these challenges. In particular, the Maasai community has encountered a multitude of barriers hindering access to essential services, including social,

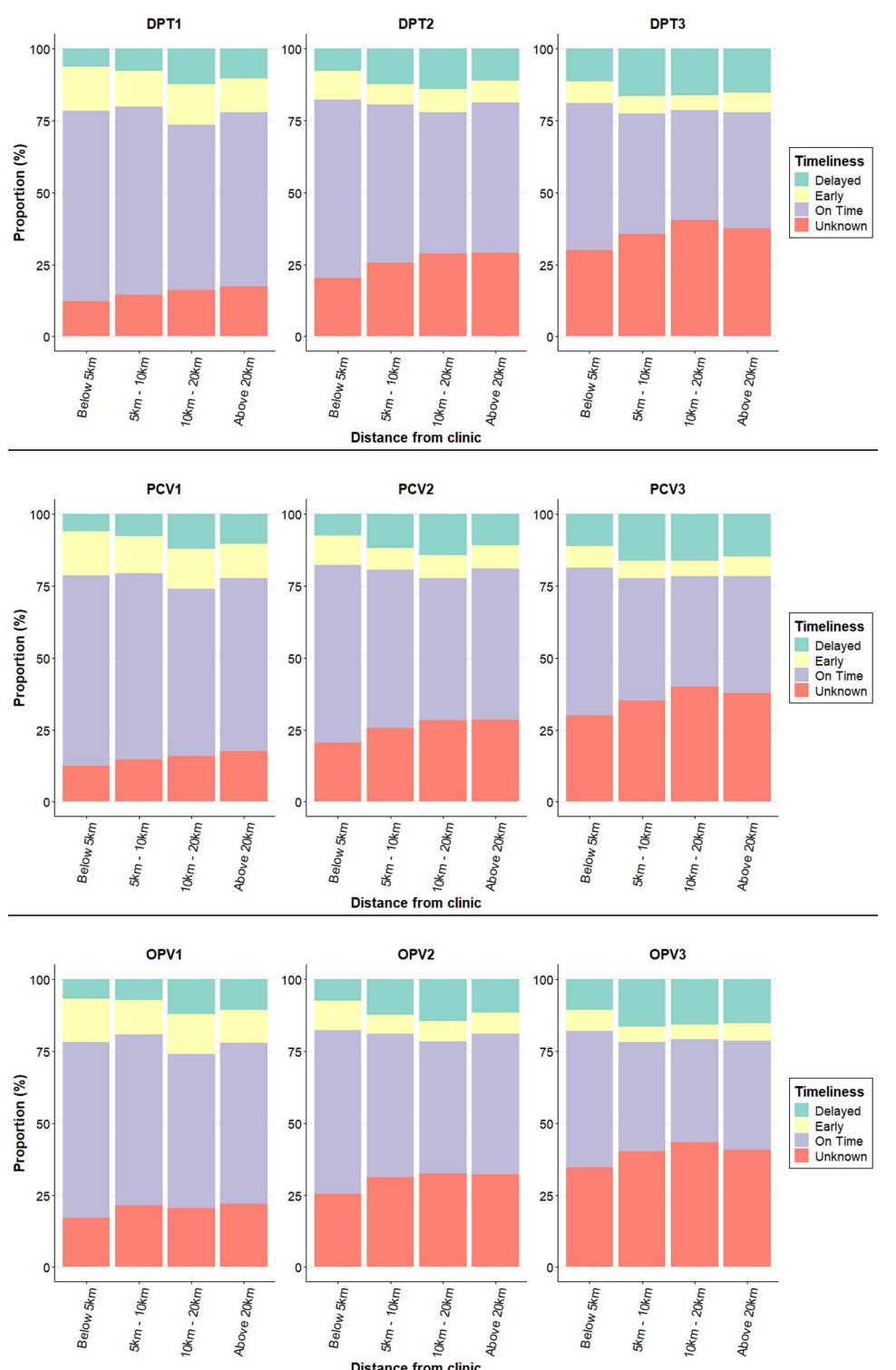

**Fig 4. Timeliness of multidose vaccines and distance from clinic.**

**Table 2.  Vaccine dropout Rate.**

|  | Dose | Overall |
|---|---|---|
| OPV | 1st-2nd | **11.41%,** 95% CI [10.66,12.18] |
|  | 2nd-3rd | **13.24%,** 95% CI [12.39,14.12] |
|  | 1st- 3rd | **23.13%,** 95% CI [22.14,24.15] |
| DPT | 1st-2nd | **11.24%,** 95% CI [10.52,11.99] |
|  | 2nd-3rd | **13.01%,** 95% CI [12.19,13.85] |
|  | 1st- 3rd | **22.78%,** 95% CI [21.82,23.77] |
| PCV | 1st-2nd | **11.26%,** 95% CI [10.54 - 12.01] |
|  | 2nd-3rd | **12.92%,** 95% CI [12.11 - 13.77] |
|  | 1st- 3rd | **22.73%,** 95% CI [21.77 - 23.71] |
| Rota | 1st-2nd | **14.70%,** 95% CI [13.83 - 15.60] |
| MR | 1st-2nd | **64.66%,** 95% CI [63.09 - 66.21] |

financial, and geographic factors, compounded by the pervasive spread of vaccine misinformation and suboptimal community engagement [5]. This aligns with similar studies in other rural settings that have reported a decline in trust towards vaccines due to beliefs that vaccines were harmful, expired and could cause disability or death among their children [26–28]. In such contexts, health workers are essential in building community trust and encouraging parents to vaccinate their children whenever the services are available.

In Kenya a child is considered fully vaccinated against all essential antigens upon receiving the BCG vaccine, three doses each of the polio vaccine (excluding the OPV at birth) and the DPT-containing vaccine, along with a single dose of the measles-containing vaccine. Nationally, the basic antigen vaccination coverage is 80% compared to 75% in Narok County (KDHS, 2022). For multi-dose vaccines, a significant drop in uptake between the second and the third doses (DPT, OPV and PCV) was observed. This decline may indicate obstacles within the healthcare system that deter parents or caregivers from returning, inadequate communication about the importance of follow-up visits, or gaps in the tracking of children registered at healthcare facilities. For example, the first dose of the MR vaccine is administered at 9 months, and the second dose at 18 months. This significantly longer interval, compared to other vaccines, likely contributes to the high rate of missed appointments for the second MR dose. Studies on vaccination dropout determinants highlight the crucial role of healthcare counseling in reducing dropout rates by effectively disseminating immunization information. Additionally, missed opportunities for administering all scheduled vaccines during a single visit, known as "non-simultaneous vaccination," can also lead to higher dropout rates. In rural populations, this issue may be due to vaccine supply shortages, errors in identifying the vaccines that are due, reluctance to vaccinate an ill child, or hesitancy to administer multiple vaccines during a single appointment. Catch-up campaigns aimed at children who miss vaccine doses should be prioritized.

Our findings also highlight the vulnerability of children living farther from vaccination-providing health facilities, as the proportion of children receiving vaccines on schedule decreases with increasing distance to these facilities. These highlight that rural populations are diverse and that inequities extend beyond the basic urban-rural divide. While geographic distance is just one of many barriers that make a population hard to reach or vaccinate, it is relatively straightforward to define and address, making it a meaningful indicator of timely immunization. However, accessibility to healthcare is influenced by multiple factors beyond distance alone. Factors such as road quality, availability of transport options, and physical or man made barriers (e.g., rivers, mountains) also play a crucial role in determining access.

These elements can significantly impact travel time and the overall feasibility of reaching healthcare facilities. Therefore, equity-focused interventions and monitoring efforts should incorporate the most granular geographic delineations possible, prioritizing children in the most remote and underserved areas [29].

The impact of increasingly frequent extreme weather events on vaccination timeliness remains understudied, despite their disruptive effects on population movement, which subsequently impacts vaccination schedules. Most recently, a retrospective study in sub-Saharan Africa observed that drought was associated with lower odds of timely completion of BCG, DPT, and polio vaccination on time [30]. Drought can negatively impact agricultural and crop production, leading to food insecurity and financial instability. With limited financial and food resources, parents may find it more difficult to afford healthcare or pay for transportation to clinics for vaccinations [31,32]. Furthermore, families may be less able to access healthcare in new geographical settings, as extreme weather conditions detrimentally impact the healthcare system and infrastructure [33].

Other barriers to timely vaccinations included inconvenient vaccination times. In Narok vaccines were typically administered only on market days and weekends to avoid wasting doses for the reconstituted vaccines, if only one child showed up. Lengthy waiting periods in Ethiopia have also been identified as a significant barrier to timely vaccinations [34,35]. Studies also highlight instances where parents forget their children's appointment dates or vaccination schedules for the next immunization visit [36,37].

With more children missing out on important vaccines each year [23], supply-side issues, such as inconsistent vaccine deliveries from the national government, can significantly derail catch-up efforts for the zero dose children. It is recommended that a country maintains at least an eight-month stock of each vaccine in its supply chain. Recently, In 2024, Kenya missed out on a year-long supply of vaccines after defaulting with a global supplier. The nationwide shortage was linked to debts owed by the government to UNICEF and the Global Vaccine Alliance, which procure and distribute vaccines through a co-funding model [38]. Furthermore, as Kenya transitions to a middle-income nation, many international donors have progressively scaled down their assistance in the recent years. This has led to limited access to routine vaccinations, leaving children more vulnerable to preventable diseases and jeopardizing their long-term health outcomes.

## Strength and limitations

Previous vaccination studies on nomadic pastoralists have focused on coverage or completion rates, which present an accumulation of the required number of doses by infants irrespective of the timing of vaccine administration. Variations in care-seeking locations and behavior can lead to inaccuracies in coverage estimates when using routine facility data. It is common for individuals to seek vaccination services outside their village, potentially resulting in an over count or under count in the numerator and misalignment between the population in need of vaccination and those receiving it within a given village. One of the key strengths of our study was its community-based approach, which allowed us to identify missed opportunities at the community level, making our findings more representative when compared to studies done at the national and sub-national level. Our findings also provide valuable insights for policy makers within the maternal health continuum on the need to implement community-level electronic immunization registries. Such systems would enhance programmatic decision-making at all levels of the health system by enabling more granular analyses and performance measurements.

Most studies predominantly rely on survey-based coverage estimates, whereas our focus on timeliness and drop-out rates provides a more comprehensive vaccination profile at the community level. However, our estimates calculated may be biased due to inaccuracies in the

numerators and denominators caused by challenges with record-keeping, failure to account for migrations and variations within the health facility catchments, and non-standardized village spelling. Record-keeping was compromised by a mix of systemic and random errors. Some paper records were often smudged, damaged, or illegible due to poor handwriting, rendering them unusable. Records were frequently crossed out and rewritten if a staff member believed the immunization date in the records did not match the date in the child's mother-child health book or if the dates appeared nonsensical. A notable systematic error was observed in recording the third dose of the Rota vaccine. The older Ministry of Health child immunization record books only had columns for two doses, while some facilities administered three. Consequently, staff sometimes recorded the third dose of Rota in the Vitamin A column, potentially compromising the reliability of the Vitamin A data. Additionally, village names were misspelt by staff, likely due to variations in regional pronunciations or record keepers spelling unfamiliar village names to the best of their ability. This inconsistency resulted in the absence of a definitive list of villages, making it challenging to accurately geo-locate them. Implementing standardized spelling of village names at each facility could alleviate this issue.

## Conclusion

Our study revealed high levels of missed opportunities in the administration of routine childhood vaccinations. A considerable number of children were not fully immunized by the end of their first year of life; even when they are fully immunized, a sizeable number received their vaccines inappropriately, either early, delayed or in a different sequence from the recommended schedule. New strategies are needed to enable health care providers and parents/guardians to work together to increase the levels of completion of all required vaccines. As Kenya seeks to build its internal self-reliance through local manufacturing, more focus is needed on measles and vaccines given in multiple doses (polio, pentavalent and pneumococcal conjugate vaccine) to make sure children receive all the doses. Overall, our study underscores the importance of community health information systems for understanding the important vaccination metrics for a nomadic community in rural Kenya. The systematic assessment of data quality and transparent adjustments are critical steps towards improving the quality of estimates derived from community health systems. Future studies should examine the unique maternal, and demographic factors associated with vaccine timeliness for nomadic communities in similar settings.

## Supporting information

**S1 Fig. Vaccination coverage by year.**
(TIFF)

**S1 Table. Timeliness of multidose vaccines.**
(DOCX)

**S2 Table. Distribution of vaccinations by facility.**
(DOCX)

**S3 Table. Vaccination counts by facility and gender.**
(DOCX)

**S4 Table. Dropout rate by dosage and facility.**
(DOCX)

**S1 Data. Vaccination dataset.**
(CSV)

## Acknowledgments

We would like to thank all community health workers for supporting our study. Furthermore, we would like to thank all community health partners facilities for providing access to immunization records as well as the Ignite Global Health Lab past summer fellows for digitizing the paper records.

## Author contributions

**Conceptualization:** Julius Nyerere Odhiambo, Carrie B Dolan, John Sankok.

**Data curation:** Julius Nyerere Odhiambo, Evelyn Thompson, Katie O'Neill.

**Formal analysis:** Julius Nyerere Odhiambo, Katie O'Neill.

**Funding acquisition:** Carrie B Dolan.

**Investigation:** Julius Nyerere Odhiambo.

**Methodology:** Julius Nyerere Odhiambo.

**Project administration:** Carrie B Dolan, John Sankok, Rose Kimani.

**Supervision:** Julius Nyerere Odhiambo, Carrie B Dolan.

**Validation:** Evelyn Thompson.

**Writing – original draft:** Julius Nyerere Odhiambo, Evelyn Thompson, Katie O'Neill.

**Writing – review & editing:** Julius Nyerere Odhiambo, Carrie B Dolan, John Sankok, Rose Kimani.

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
