## [Decision Letter · Decision Letter 0]

10 Sep 2024

PGPH-D-24-01546

Childhood vaccination trends among the Maasai nomadic pastoralists: Insights from a community vaccine registry in Kenya

Dear Dr. Odhiambo,

Thank you for submitting your manuscript to PLOS Global Public Health. After careful consideration, we feel that it has merit but does not fully meet PLOS Global Public Health’s publication criteria as it currently stands. Therefore, we invite you to submit a revised version of the manuscript that addresses the points raised during the review process.

Please ensure clarity and consistency of your results and methods, both in the body of the paper and in the abstract. Additionally, reviewer 2 makes an important point about more appropriately contextualising your study population and setting. This is needed to provide a sound base for the importance of your study.

Regarding methods and results, we anticipate that most points will be able to be sufficiently addressed through revision. Where this is not possible, please provide clear rationale as to why the point was not able to be addressed.

Regarding data availability, please review the PLOS GPH guidelines https://journals.plos.org/globalpublichealth/s/data-availability and ensure that these are sufficiently met with an accompanying clear explanation.

Best Wishes,

Gemma

We look forward to receiving your revised manuscript.

Kind regards,

Gemma Lea Saravanos

Academic Editor

Journal Requirements:

2. We have amended your Competing Interest statement to comply with journal style. We kindly ask that you double check the statement and let us know if anything is incorrect. 

3. Please provide separate figure files in .tif or .eps format.

4. Figure 1 and 3: please (a) provide a direct link to the base layer of the map (i.e., the country or region border shape) and ensure this is also included in the figure legend; and (b) provide a link to the terms of use / license information for the base layer image or shapefile. We cannot publish proprietary or copyrighted maps (e.g. Google Maps, Mapquest) and the terms of use for your map base layer must be compatible with our CC-BY 4.0 license. 

Additional Editor Comments (if provided):

Reviewers' comments:

Reviewer's Responses to Questions

**Comments to the Author**

1. Does this manuscript meet PLOS Global Public Health’s publication criteria ? Is the manuscript technically sound, and do the data support the conclusions? The manuscript must describe methodologically and ethically rigorous research with conclusions that are appropriately drawn based on the data presented.

Reviewer #1: Yes

Reviewer #2: Yes

2. Has the statistical analysis been performed appropriately and rigorously?

Reviewer #1: Yes

Reviewer #2: I don't know

3. Have the authors made all data underlying the findings in their manuscript fully available (please refer to the Data Availability Statement at the start of the manuscript PDF file)?

Reviewer #1: Yes

Reviewer #2: No

4. Is the manuscript presented in an intelligible fashion and written in standard English?

Reviewer #1: Yes

Reviewer #2: Yes

5. Review Comments to the Author

Reviewer #1: A well written paper that contributes to framing a more comprehensive profile of Kenya's routine immunization coverage, timeliness and dropout rates.

A few of the sentences could benefit from being rewritten for clarity. An example can be found in line 325 in the sentence starting with 'With limited access to routine vaccines, children become more vulnerable...'

Another example is from line 172 to 174. While a trained person would know that '...the difference between consecutive doses...' is referring to *the difference between children receiving consecutive doses*, this can be stated to aid clarity.

Reviewer #2: Thank you for this interesting article about an important topic in a minority population. I hope that the following suggestions would help to refine this paper.

General:

- Title: Subtle change, but I would call it “community-based” - given that as far as I have understood from your manuscript, the process of keeping local vaccination records is national policy, using a national template. It is just that individual sites do not share their data nationally. If I have misunderstood this, then it should be better clarified in the article.

- Data sharing: There is a note to say that the data will be made available with an email address to contact - but I do not believe this is allowed under the journal policies.

- Appropriateness of statistical analyses: Not enough detail has been given in some of the analyses for me to have a clear idea of whether the methods were appropriate.

Abstract:

- The number of villages in the abstract is different from the number in the main results section of the paper.

- The headline results presented in the abstract are not presented in the main results section of the paper.

Background:

The background section currently is around 85% about the global childhood vaccination situation, and only 15% about what is happening in Kenya (with nothing in the background about Narok). To be an effective paper, it really needs to be the other way around, with a real emphasis on the communities you are studying. You really need to give a lot more context specific to the local area, as well as the rationale for undertaking this particular study (in this particular area and with these particular clinics) and its anticipated contribution. The word “Maasai” was not even mentioned once in the background section!

- Please give us much more detail on the population within this study area! For example, are they are homogenous ethnic group (all Maasai) and what is the age distribution? Is there any indication of the size of the annual birth cohort?

- The words “nomadic pastoralists” was mentioned in the title, but not anywhere else in the article! Readers are from around the world, where there many different kinds of nomadic pastoral lifestyles - therefore there needs to be better description of what this term means in the context of the Maasai. What specific challenges does this pose for childhood immunisation?

- You were discussing access to clinics only in terms of distance - how about other factors like quality of roads, transport options, any specific physical or manmade barriers to travel?

- Are there any costs associated with childhood vaccines? Even if the vaccines themselves are free, do parents need to pay a small service fee to be seen at the clinic?

- Are these clinics the only way for children to receive childhood vaccines? Or are there community healthcare workers who visit door to door?

- Was this analysis part of a bigger project on improving immunisation coverage in this region? The context of data collection for this study could also be clarified.

- It would be good to know more about the training received by the clinic staff - you had mentioned that there were errors in record keeping.

-- Are clinical staff trained to calculate a catch-up schedule for children who are delayed with their vaccines? Are there any guidelines about catching children up?

- Where you’ve said “most prevalent in the first year of life” - firstly, “prevalent” is probably not the right word to use here, but difficult to know without knowing what evidence this sentence is based on. Do you have evidence to support this sentence?

Methods:

(I fully recognise that it would take a lot of words to answer some of the following questions - for the sake of the word count, maybe it would be useful to include some of these explanations in the supplementary information.)

- Good use of subheadings to help readers keep track of the different considerations that went into the design of this study.

- While you have given the dates when you were extracting records, you have not given the exact period that the records date from. Do we assume that these are the complete immunisation records of each clinic dating back to when they first started record keeping until a certain date in 2024?

- Have you left a gap in terms of the period of analysis and the period of data availability? In other words, if you collected data until June 2024, then do you analyse data on vaccine doses due until maybe March 2024, so that you have 3 months to see if the child will be late or might drop out altogether?

- You mentioned there were errors or other quality issues in the record keeping. Was there a checklist / a list of criteria that you used to judge the quality of a record - and did you exclude any records that did not meet minimum data quality standards?

- Digitalisation of the data collected was mentioned - how exactly was this done? Were paper records just scanned into digital images? Was the data fed into a study database? Does this digitalised data get shared back to these community clinics?

- The serial number assigned to each child, is this number used across the 4 clinics? In other words, if a child moves from one area to another and receives their vaccines from two different clinics within the 4 that are included in the study, could you track this child from one clinic to the next based on their serial number?

- Where are the date of birth and other personal details obtained from - directly from the mother and child booklet?

- How reliable are the dates of birth - given that they are the basis of many of the calculations in this paper? Were there records where the date of birth was missing, and what was done with these records? Were they excluded from analysis?

- Did the analysis of timeliness take into account the interval after the previous dose? So for example, according to your table, somebody could get PCV2 at 97 days of age and be counted as “on time”, but then they can get PCV3 at 99 days of age and be counted as “on time” again. But an interval of 2 days between these 2 doses would definitely not produce the optimal immune response! This would be a lot of work to review the timing of vaccine doses for each individual child, so it is perfectly reasonable if this degree of analysis was not done - but it needs to be mentioned as a limitation to the study.

-- Similarly, do you account for any catch-up timetables for children with delayed vaccination?

- How did you deal with any discrepancies between what was in the clinic immunisation register and the mother-and-child booklet? Which of these was used as the “source of truth” where there are discrepancies?

- What was the method used to obtain 95% confidence intervals?

- What are the date criteria for MR dose 1 and 2 in terms of the number of allowable days before it counts as “delayed”? Those cells in the table are blank.

Results

- Four primary healthcare clinics: are they they only primary healthcare clinics in this region? What is the population size that they serve? What proportion of this population are children? What are the difference between these clinics - if any? What are the differences in the characteristics of the people served by each clinic? Is the patient base of one clinic more nomadic/transient than others?

- With the 8487 records, is each record 1 child (with all their vaccines included)? Or is each record 1 line in the immunisation register (in which case each child may have more than 1 record)? This really needs to be made clear.

- It would be useful to have a descriptive statistical analysis of data obtained from each of the 4 clinics: how many records from each clinic? How many vaccine doses were given at each clinic?

- Figure 2 shows a sharp but transient improvement in 3rd dose coverage for all 3 vaccines in year 2020. Why was this? The paper should explain.

- Figure 3:

-- The 1st paragraph of results said that there were 258 villages represented in the data. However, there are clearly fewer than 258 individual dots in each of the 6 maps. It would be useful to include a footnote explaining which villages were excluded from the maps - was it because there was insufficient data from the village? And if so, what was the cutoff used to decide whether a village had enough data to display?

-- The locations of the community clinics are not displayed on the map - therefore readers cannot see the relationship between distance from the clinic with any outcomes.

- Vaccine coverage by village: if I have understood correctly, then there are 8487 children from 258 villages. This means that there are on average 33 children per village. And then this number is sub-divided by year when you calculate percent coverage each year, which means that on average there are probably fewer than 10 children per village who are due for this particular antigen and could be counted in the denominator. It is not appropriate to calculate percentage coverage with such a small denominator, or to draw conclusions at the village level. I would recommend working with a larger geographical unit - if possible, so that there are at least 100 individuals in the denominator.

- Timeliness: you had said the *highest* decline between the first and 3rd doses was observed in DPT, PCV and OPV - but given that these are the only 3-dose vaccines that are included in this study, the word “highest” should be removed.

- Figure 4: Are there equal numbers of children in each category? It is difficult to be able to draw conclusions/exclude statistical artefacts from this figure without knowing whether there may be hundreds of children in the category closest to the clinic but only dozens of children in the furthest category

- Were the children in the “delayed” group mutually exclusive from children in the “dropout” group? So you waited for a certain length of time after the vaccine was due, and if they had not received the vaccine by then, they counted as “dropout”? The methodology of classifying children as “delay” or “dropout” needs to be clear.

- You have not made any comments on zero-dose (beyond initial BCG) children. What percentage of children received a BCG vaccine as part of their “enrolment” in the clinic denominator, but then never received any subsequent vaccines?

Discussion:

- You have recommended to use “the most granular geographic delineations possible.” Would knowing somebody’s address to the nearest 100m result in a significant difference to public health measures compared to knowing their address to the nearest 1km?

- Drought is described as a potential contributor to vaccination delays, with evidence that this is the case in sub-Saharan Africa. However, you have not linked this to the specific communities you are describing. Did they suffer droughts during the period of time that your analysis covered? And if so, did it affect the entire community equally, or was it more obvious in the records of one clinic?

- The results (especially the supplementary material) have shown that many doses of vaccines are given early - especially the vaccines at the 6-week-old time point, where more vaccines are given early than late. However, this is not mentioned at all in the discussions. This is quite a glaring omission, and it would be useful to know why so many doses of vaccines might be given early.

- Are there any particular health impacts observed in Narok that might be due to a lower childhood vaccination coverage compared to the rest of the country?

- You had mentioned supply issues - were there any specific interruptions to supply during the time covered by the study? And if yes, was there an observable impact on local vaccine coverage at these 4 clinics?

- You mentioned transition out of Gavi support - it might be useful to give the reader an idea about where Kenya is exactly in this trajectory, and the feasibility of adequate domestic manufacturing.

- You may want to shift the recommendation for electronic immunisation records until later in the discussion section. At the moment, it comes across as a very sudden idea that is not linked to any other ideas - and it is only until later, when you discuss data entry errors or poor legibility that it becomes obvious why you are suggesting this. It might be useful to move the recommendation to use electronic records after the explanation about poor data quality.

Conclusion:

- Succinct conclusion, no specific comments other than that it might be useful to think about how the results of this study can be used to train vaccination providers at these clinics.

Overall:

- Presenting the data only as proportions and not as absolute counts means that the reader cannot fully interpret the data that they are seeing. It would be very helpful for your paper to have data presented both ways.

6. PLOS authors have the option to publish the peer review history of their article (what does this mean? ). If published, this will include your full peer review and any attached files.

**Do you want your identity to be public for this peer review?** For information about this choice, including consent withdrawal, please see our Privacy Policy .

Reviewer #1: No

Reviewer #2: No

---

## [Decision Letter · Decision Letter 1]

26 Dec 2024

PGPH-D-24-01546R1

Childhood vaccination trends among the Maasai nomadic pastoralists: Insights from a community-based vaccine registry in Kenya

Dear Dr. Odhiambo,

Thank you for submitting your manuscript to PLOS Global Public Health. After careful consideration, we feel that it has merit but does not fully meet PLOS Global Public Health’s publication criteria as it currently stands. Therefore, we invite you to submit a revised version of the manuscript that addresses the points raised during the review process.

Please consider and respond to all of the reviewers comments, highlighting where changes could and could not be made including a clear rationale.

In summary, the critical changes needed include:

1. Please revise table 1 considering the reviewer comments and provide the source reference of this information as a table footnote.

2. Please revise your results section to improve the clarity of the denominator and excluded records, include additional supplementary tables where possible.

- Ensure that methods and discussion of limitations are revised accordingly.

3. Regarding the geographic unit of village, is it possible to expand this to a larger unit such as a subcounty or district? As highlighted by the reviewer, calculating coverage with such small sample sizes is not robust. If not possible, please highlight the limitations of coverage calculations at the village level in the discussion.

4. Please revise relevant parts of the discussion in particular:

- revision of the limitations section

- ensure supporting references are provided when referring to ‘previous relevant studies or literature’

- ensure supporting references for specific information such as " It is recommended that a country maintains at least an eight-month stock of each vaccine in its supply chain"

- expand on or remove reference #33 - anonymous MoH is not sufficient

5. Figure 2 is missing from the revised manuscript – please correct this.

We look forward to receiving your revised manuscript.

Kind regards,

Gemma Lea Saravanos

Academic Editor

Journal Requirements:

Reviewers' comments:

Reviewer's Responses to Questions

**Comments to the Author**

1. If the authors have adequately addressed your comments raised in a previous round of review and you feel that this manuscript is now acceptable for publication, you may indicate that here to bypass the “Comments to the Author” section, enter your conflict of interest statement in the “Confidential to Editor” section, and submit your "Accept" recommendation.

Reviewer #2: (No Response)

2. Does this manuscript meet PLOS Global Public Health’s publication criteria ? Is the manuscript technically sound, and do the data support the conclusions? The manuscript must describe methodologically and ethically rigorous research with conclusions that are appropriately drawn based on the data presented.

Reviewer #2: Yes

3. Has the statistical analysis been performed appropriately and rigorously?

Reviewer #2: I don't know

4. Have the authors made all data underlying the findings in their manuscript fully available (please refer to the Data Availability Statement at the start of the manuscript PDF file)?

Reviewer #2: Yes

5. Is the manuscript presented in an intelligible fashion and written in standard English?

Reviewer #2: Yes

6. Review Comments to the Author

Reviewer #2: (Also uploaded as Word document attachment in case formatting does not come across clearly)

Thank you for all your revisions, and for your very helpful answers to my previous questions!

However, not all the points that you had said you would address in the manuscript made their way into the final manuscript. For example, there was mention of supplementary table 3 and 4 that were meant to have been added, but did not make their way to the final manuscript. It might be useful to review the final manuscript in parallel with your responses to the reviewer to make sure that all planned edits had been incorporated.

[Editors: I just wanted to check that there weren’t additional supplementary files somewhere else? I only saw the new “immunisation_data.csv” file.]

The other major area to focus on is the denominator for the coverage estimates - making sure that this is as clearly explained and as transparently reported as possible.

Background:

Great work on your edits. This now helps me (and readers) understand the context so much better, and also enriches the paper. Thank you!

Methods:

- Thank you for clarifying that the clinic record was regarded as the “source of truth” in cases of discrepancies between the two sources - it would also be good to put this into the paper

- Table 1: Thank you for filling in the table with some date criteria for MR, but I wonder if there are errors

- The “delayed” criterion for MR1 is not possible, given that it occurs earlier than the “early” criterion

- No “delayed” criterion for MR2

- Important: the entire table needs referencing

Results:

- Thank you for including the actual dataset in the supplementary material - now I think I can see your methods for some of the data analyses that I had previously queried. I think that to make your results absolutely clear, the following points need to be included:

- State that the 8487 records is your overall denominator: Each of these records corresponds to a child who has had a dose of BCG or who has migrated into the catchment of one of these clinics, and each child may have 1 or more vaccines recorded (this was what I had wanted when I had asked for absolute clarity in my previous review about whether each record refers to 1 child or 1 vaccine dose - it makes a huge difference to your sample size!)

- State whether there was any adjustments to the denominator to account for children who migrated into the area, or whether there was just an assumption that migration in roughly equated to migration out

- Thank you for letting me know in your response that you did exclude some records due to data quality. Given that this is your denominator and is crucial to your calculation of coverage, it would be good to also include this information in the manuscript itself, and an indication of how many records were excluded from the denominator due to poor data quality

- Given that you are reporting annual coverage, you need a separate table outlining the denominator you used for each year

- Given that you had to adjust the denominator for each dose of each vaccine to account for eligibility, it would be useful to add another table to the supplementary material listing the actual denominator used to calculate coverage for each dose of each vaccine - this ensures reproducibility of your results

- It would be useful to clarify whether all children at each clinic were of Maasai background, or only a proportion

- I had previously made the comment: “Vaccine coverage by village: if I have understood correctly, then there are 8487 children from 258 villages. This means that there are on average 33 children per village. And then this number is sub-divided by year when you calculate percent coverage each year, which means that on average there are probably fewer than 10 children per village who are due for this particular antigen and could be counted in the denominator. It is not appropriate to calculate percentage coverage with such a small denominator, or to draw conclusions at the village level. I would recommend working with a larger geographical unit - if possible, so that there are at least 100 individuals in the denominator.”

- I can see that you had responded to this comment - however, I think you may not have understood my original point. I do not see problems with sample size when you look at all 8487 records as a whole. My comment only applies to Figure 3 and related analyses: when you break this number down by village AND year AND specific vaccine dose, then I doubt you have an adequate sample size to really support any firm conclusions being drawn at this village level.

- I think to keep this part of the analysis in the paper, it would be important to highlight that due to small sample sizes at the village level, this is the trend that the data appears to show

Discussion:

- The added comment about the context of supply / funding is very useful - thank you!

- It would be useful to comment on the fact that these communities show a similar trend to pretty much everywhere in the world - that coverage drops in subsequent doses!

- But this also means that a useful recommendation is to learn from other systems around the world about the strategies used to improve coverage of subsequent doses

- Limitations:

- You need to make it clear that one limitation in your analysis of geographical trends is that you have not explored whether there are other explanations for what you had observed: for example, could it be that people living in more peripheral villages have a more nomadic lifestyle than those living closer to clinic locations? Could it be that people in the peripheral areas just went over the border to receive vaccines in a county not covered by this study? Could it be that peripheral villages have a smaller population, and so the presence of one under-vaccinated child can cause a greater fluctuation in the data (and therefore the findings can be attributed to statistical artefact)?

- In other words, access to clinics may not be the ONLY reason why the coverage appears to be lower in the more distant villages - there may be a range of other factors

- Another limitation: using receipt of BCG vaccine as a denominator, which means that the children who do not receive a birth dose BCG at all are excluded from this study

- Thank you for all your responses to my questions during the previous review. Your responses also prompted the need to include another limitation: that these clinics are not the only source of vaccines in the county, and therefore there may be a small possibility that children who were recorded as having “missed” doses at these clinics may have received these doses through other channels. This issue of the clinics not capturing all doses is also very important to consider in light of describing a highly mobile population - are these children necessarily confined to this particular county with its 4 sub-county clinics for their vaccines?

- Another limitation from responses: that the study does not look at the interval between 2 individual doses in its assessment of “early” or “delayed” - as I have said previously, a reasonable decision to make based on human/time resources, but just needs to be mentioned in the limitations

Minor edits:

- Check for any unnecessary capitalisation in the manuscript - for example, “Pneumococcal Conjugate Vaccine”

Referencing:

[Editors, I just wanted to check a couple of things:

- Ref #33 - have enough details been provided in the bibliography?

- Shapefile for map: the source of the base map was given in the figure itself. Does this also need to be included in the bibliography?]

7. PLOS authors have the option to publish the peer review history of their article (what does this mean? ). If published, this will include your full peer review and any attached files.

**Do you want your identity to be public for this peer review?** For information about this choice, including consent withdrawal, please see our Privacy Policy .

Reviewer #2: No

---

## [Editor Report · Decision Letter 2]

5 Feb 2025

Childhood vaccination trends among the Maasai nomadic pastoralists: Insights from a community-based vaccine registry in Kenya

PGPH-D-24-01546R2

Dear Dr Odhiambo,

We are pleased to inform you that your manuscript 'Childhood vaccination trends among the Maasai nomadic pastoralists: Insights from a community-based vaccine registry in Kenya' has been provisionally accepted for publication in PLOS Global Public Health.

Best regards,

Gemma Lea Saravanos

Academic Editor